# In-Silico Functional Metabolic Pathways Associated to *Chlamydia trachomatis* Genital Infection

**DOI:** 10.3390/ijms232415847

**Published:** 2022-12-13

**Authors:** Simone Filardo, Marisa Di Pietro, Marta De Angelis, Gabriella Brandolino, Maria Grazia Porpora, Rosa Sessa

**Affiliations:** 1Department of Public Health and Infectious Diseases, Microbiology Section, University of Rome “Sapienza”, 00185 Rome, Italy; 2Department of Maternal and Child Health and Urology, University of Rome “Sapienza”, 00185 Rome, Italy

**Keywords:** *Chlamydia trachomatis*, cervicovaginal microbiota, metagenomic data, in-silico metabolic profiling, PICRUSt2

## Abstract

The advent of high-throughput technologies, such as 16s rDNA sequencing, has significantly contributed to expanding our knowledge of the microbiota composition of the genital tract during infections such as *Chlamydia trachomatis*. The growing body of metagenomic data can be further exploited to provide a functional characterization of microbial communities via several powerful computational approaches. Therefore, in this study, we investigated the predicted metabolic pathways of the cervicovaginal microbiota associated with *C. trachomatis* genital infection in relation to the different Community State Types (CSTs), via PICRUSt2 analysis. Our results showed a more rich and diverse mix of predicted metabolic pathways in women with a CST-IV microbiota as compared to all the other CSTs, independently from infection status. *C. trachomatis* genital infection further modified the metabolic profiles in women with a CST-IV microbiota and was characterized by increased prevalence of the pathways for the biosynthesis of precursor metabolites and energy, biogenic amino-acids, nucleotides, and tetrahydrofolate. Overall, predicted metabolic pathways might represent the starting point for more precisely designed future metabolomic studies, aiming to investigate the actual metabolic pathways characterizing *C. trachomatis* genital infection in the cervicovaginal microenvironment.

## 1. Introduction

The female genital tract is an ecological niche harboring the simplest commensal bacterial community of the human body [1,2,3]. Indeed, cervicovaginal microbiota from the majority of healthy reproductive-age women is most often characterized by reduced bacterial diversity and the dominance of *Lactobacillus* spp. [4].

During the years, the *Lactobacillus* spp. dominated cervicovaginal microbiota has been recognized as an important host defense factor against different pathogens, hindering their invasion through different mechanisms, including the acidic pH [5,6,7]. Indeed, following the depletion of lactobacilli, an overgrowth of diverse anaerobic bacterial species with an increase in pH occurs, a condition generally called dysbiosis, usually associated with an increasing risk for acquiring different pathogens, such as *Chlamydia trachomatis* [4,8].

*C. trachomatis* is recognized as the most common sexually transmitted bacterial infection and is characterized by a high proportion of asymptomatic infections in women [9]. Indeed, approximately 80% of women with *C. trachomatis* genital infection remain undiagnosed and untreated, potentially leading to severe reproductive sequelae, such as pelvic inflammatory disease, ectopic pregnancy, and obstructive infertility [9,10].

The advent of high-throughput technologies such as 16s ribosomal RNA encoding gene sequencing has significantly contributed to expanding our knowledge on the composition of genital microbiota, identifying different bacterial profiles, that have been classified into five community state types (CSTs) [11,12]. Specifically, CST-I, -II, -III, and -V are dominated by *Lactobacillus crispatus*, *Lactobacillus gasseri*, *Lactobacillus iners*, and *Lactobacillus jensenii*, respectively, whereas CST-IV is characterized by a reduced number of *Lactobacillus* spp. and an increased diversity of anaerobic bacterial species, including *Gardnerella, vaginalis*, *Atopobium vaginae*, *Prevotella* spp., etc. [12]. In this regard, a healthy genital condition is generally associated with an *L. crispatus* (CST-I) and *L. gasseri* (CST-II) dominated microbiota [6,13]; by contrast, women with *C. trachomatis* infection possess a genital microbiota dominated by *L. iners* (CST-III) or by different anaerobic bacteria (CST-IV) [2,8,14,15].

Beside specific microorganisms, other factors, such as their metabolites, may play a role in preventing or favoring the invasion and intracellular replication of *C. trachomatis* [16,17]. It is indeed known that *L. crispatus* and *L. gasseri* produce lactic acid, responsible for the low vaginal pH, one of the main defense mechanisms of the host against invading pathogens [18,19], although the role of other metabolic products in different cervicovaginal microbial communities still need to be investigated.

As a result, a clear picture of the metabolic roles of each microorganism within a microbial community and their metabolite-based interactions involved in health and disease is of great pathological importance. In this regard, functional characterization of the microorganisms that colonize the cervicovaginal environment may be helpful to better understand how their metabolites can impact the ability of *C. trachomatis* to compete and colonize the genital tract. To reach this goal, several powerful computational approaches for the in-silico functional analysis, such as PICRUSt2 and HUMAnN 2.0, able to predict the metabolic pathways from metagenomic data, have been introduced [20,21].

Therefore, in this study, we investigated the predicted genetic metabolic pathways, via PICRUSt 2 algorithm, from 16s rDNA sequence data of microbial communities associated with *C. trachomatis* cervicovaginal infections, in relation to the different CSTs.

## 2. Results

### 2.1. Metagenomic and Functional Metabolomic Profiling of 16S rDNA Dataset

An average of 74,122 [median (IQR) 82,196 (32,576)] and 65,341 [median (IQR) 75,850 (54,769)] paired-end Illumina reads were analyzed per sample in healthy controls and *C. trachomatis* positive women, respectively. Reads with a quality score of <20 were excluded from downstream analysis.

The cervical microbiota from healthy controls was characterized by the highest relative abundance of *Lactobacillus crispatus* (mean relative abundance, 45.8% [standard error of the mean {SEM}, 4.5]), followed by *L. iners* (18.1% [33.5]) and *L. gasseri* (17.9% [3.1]). By contrast, a decrease in *L. crispatus* (17.6% [5.1]; *p*  <  0.01) and *L. gasseri* (0.9% [0.9]; *p*  <  0.001) was described in the cervical microbiota of *C. trachomatis*-positive women. Their cervical microbiota was characterized, indeed, by a higher prevalence of *L. iners* (32.6% [6]; *p*  <  0.0001) alongside anaerobic bacteria, such as *Gardnerella vaginalis* (15.5% [3.5] versus 5.5% [1.8]; *p*  <  0.000001), *Prevotella amnii* (3.2% [1.2] versus 0.001% [0.001]; *p*  <  0.05), *Prevotella buccalis* (1.7% [0.7] versus 0.004% [0.002]; *p*  <  0.01), *Prevotella timonensis* (1.6% [0.7] versus 0.05% [0.04]; *p*  <  0.01) and *Aerococcus christensenii* (1% [0.7] versus 0.001% [0.001]; *p*  <  0.0001) (Appendix A).

The microbial community of each *C. trachomatis*-positive woman and healthy control was assigned to its specific CST, showing most *C. trachomatis*-positive women belonging to either CST-III (predominance of *L. iners*, 35.89%) or CST-IV (predominance of mixed anaerobic flora, 38.46%). By contrast, healthy controls were mostly assigned to CST-I (predominance of *L. crispatus*, 56.57%), CST-II (predominance of *L. gasseri*, 17.17%), and CST-III (predominance of *L. iners*, 20.20%), whereas only a minority belonged to the CST-IV (predominance of mixed anaerobic flora, 6.06%) (Appendix A). More details concerning the characteristics of the cervicovaginal microbiota are reported in our previous paper [2].

### 2.2. Alpha- and Beta- Diversity Analysis of MetaCyc Pathways

The in-silico predictions of the metabolic pathways were retrieved via the PICRUSt2 bioinformatic algorithm, and, to compare the richness and diversity of predicted metabolic profiles between *C. trachomatis*-positive women and healthy controls, alpha- and beta-diversity analyses were performed.

Alpha-diversity measures (Shannon’s and Pielou’s evenness diversity indexes) showed a statistically significant increase in the metabolic pathways predicted in the cervicovaginal bacterial communities of *C. trachomatis*-positive women as compared to those in healthy controls (Figure 1A, *p* < 0.01). Bray–Curtis index, a measure of beta diversity, also revealed a statistically significant separation between the metabolic pathways of the cervicovaginal microbiota in the two study groups (Figure 1B, *p* = 0.001).

In order to investigate the contribution of the different predominant bacterial species to the predicted metabolic pathways, we analyzed the metabolic profiles of the CSTs in the cervicovaginal microbiota of *C. trachomatis*-positive women and healthy controls. In this regard, alpha-diversity measures showed a statistically significant increase in the metabolic pathways predicted in a CST-IV microbiota as compared to those in all the other CSTs (Figure 2A,B, *p* < 0.05), in either *C. trachomatis*-positive women or healthy controls. Similarly, Bray–Curtis index evidenced a statistically significant separation between the metabolic pathways of women with a CST-IV microbiota and those of all the other CSTs, in both study groups (Figure 2A,B, *p* = 0.001). Lastly, we compared the alpha- and beta-diversity measures of the predicted metabolic pathways between the CST-IV microbiota from *C. trachomatis*-positive women and those from healthy controls, to assess whether *C. trachomatis* genital infection induced further alterations. Both Shannon’s and Pielou’s evenness diversity indexes evidenced a statistically significant increase in the metabolic pathways in the CST-IV microbiota from *C. trachomatis*-positive women as compared to those from healthy controls (Figure 3A, *p* < 0.05). Also, the Bray–Curtis index confirmed the statistically significant separation of the metabolic pathways predicted in the CST-IV microbiota from the two study groups (Figure 3B, *p* = 0.001).

### 2.3. Composition of Predicted MetaCyc Pathways

Statistically significant differences were observed in the composition of predicted metabolic pathways between *C. trachomatis*-positive women and healthy controls, as evidenced in Figure 4. Most differences were associated with the presence of *C. trachomatis* genital infection and were characterized by the prevalence of metabolic pathways belonging, mostly, to the pathways for the generation of precursor metabolites, including some amino acids such as arginine, L-methionine and L-tyrosine, and energy production, such as glycolysis and tricarboxylic acid (TCA) cycle pathways, as well as for vitamin and co-factor biosynthesis, including tetrahydrofolate. By contrast, healthy controls differed in the prevalence of metabolic pathways belonging to the super-pathways for the degradation of carbohydrates.

As for the composition of predicted metabolic pathways associated with the CSTs, the only statistically significant differences were observed when comparing the metabolic pathways in the CST-IV microbiota between *C. trachomatis*-positive women and healthy controls, as evidenced in Figure 5. All differences in the predicted metabolic pathways were associated with the presence of *C. trachomatis* genital infection and were characterized by the prevalence of the pathways concerning the biosynthesis of nucleotides, amino-acids (L-lysin, L-aspartate, and L-asparagine), co-factors, such as coenzyme-A (CoA), and tetrahydrofolate. By contrast, no differences were observed when comparing the predicted metabolic pathways amongst CST-I, CST-II, and CST-III microbiota, in either *C. trachomatis*-positive women or healthy controls, or when each CST was compared between the two study groups.

## 3. Discussion

In recent years, advanced in-silico approaches, investigating the growing body of 16s rDNA sequences from microbial communities, have acquired importance for further disclosing additional information lying beneath the surface of metagenomic data, as it has been already evidenced in the human gut microbiome [22]. In our study, the predicted metabolic pathways associated to *C. trachomatis* genital infection were investigated via PICRUSt2 algorithm, adjusting for the different CSTs.

*C. trachomatis*-positive women enrolled in the present study mostly showed a CST-IV microbiota, followed by a CST-III and a CST-I microbiota, whereas the majority of healthy women possessed a CST-I, followed by a CST-II and a CST-III microbiota, with only a small percentage characterized by a CST-IV microbiota [2]. The main result of our computational analysis is the observation of a more rich and more diverse mix of predicted metabolic pathways in women with *C. trachomatis* genital infection as compared to healthy controls, as evidenced by increased alpha- and beta-diversity indexes. Specifically, a mixed anaerobic flora (CST-IV microbiota) concurred in altering the predicted metabolic pathways independently from *C. trachomatis* infection. By contrast, a CST-I, CST-II, or CST-III microbiota, characterized by a Lactobacillus spp. dominated cervicovaginal micro-environment, did not lead to a significant modification in the metabolic pathways. These findings are particularly interesting, and hint at the CST-IV microbiota, a condition of dysbiosis frequently observed in bacterial vaginosis [12], as the main factor contributing to reshaping the metabolic pathway profile, potentially favoring *C. trachomatis* infection. Specifically, a CST-IV microbiota was associated with an increased prevalence of the metabolic pathways known to support chlamydial intracellular growth and survival [16,17,23], including the energy production, such as glycolysis and the tricarboxylic acid (TCA) cycle, and the biosynthesis of biogenic amines (polyamines), carbohydrates (3-deoxy-D-manno-octulosonate), nucleotides (adenine, adenosine, pyrimidine, etc.), tetrahydrofolate, and precursor metabolites, such as CoA, that is involved in the TCA cycle or in mixed acid fermentation [24,25]. Furthermore, *C. trachomatis* is highly dependent on the metabolism of the amino-acid tryptophan, which participates in the tetrahydrofolate pathway by providing one-carbon groups for nucleotide biosynthesis, as well as of other amino acids and different cofactors, due to its limited biosynthetic capabilities [23,26].

More interestingly, in our study, *C. trachomatis*, within a cervical microenvironment with a CST-IV microbiota, was able to further modify the already altered metabolic pathways. Indeed, all the predicted metabolic pathways were present in even higher abundances in women with CST-IV microbiota and *C. trachomatis* genital infection. A similar profile associated with chlamydial infection was observed in a previous metagenomic study via a different computational approach (HUMAnN2 v3.0), further supporting the robustness of in-silico analyses [24].

The biological significance of predicted metabolic profiles observed in our study is strengthened by metabolomic studies investigating the cervicovaginal metabolic signatures in the context of *C. trachomatis* genital infection. The most recent study found lower levels of several carbohydrates and derivatives, as well as of all the metabolites involved in the TCA cycle, via untargeted gas chromatography-mass spectrometry (GC-MS) [16]. Moreover, increased levels of biogenic amines such as putrescine and cadaverine, as well as multiple long-chain fatty acids, were observed in women with a CST-IV microbiota and *C. trachomatis* infection [16]. By contrast, Parolin et al., 2018, found similar bacterial and metabolic signatures between *C. trachomatis* infected and healthy women, observing, instead, the most significant changes across multiple metabolic pathways in women with bacterial vaginosis alone, via targeted proton nuclear magnetic resonance (1H-NMR); this might be attributed to the absence of bacterial vaginosis in the women positive to *C. trachomatis* infection [17]. Our predicted metabolic pathways, indeed, are in accordance with the metabolic findings by Borgogna et al. and Parolin et al., pointing to the role of CST IV microbiota in favoring the growth and survival of *C. trachomatis*.

Overall, our study highlighted the importance of computational approaches for shedding light on further aspects that might be hidden within the routine bioinformatic analysis of metagenomic data. Indeed, PICRUSt2 proved to be a valuable and powerful tool for predicting metabolic pathways from 16s rDNA metagenomic data, complementing the microbial information, and potentially leading to deeper insights into the functional relationship of the different bacterial species inhabiting the cervicovaginal microbiota during *C. trachomatis* infection. There are, however, some limitations inherent to the usage of in-silico analyses, including the bias towards existing reference genomes, leading to underestimating rare environment-specific functions, and the difficulties of amplicon-based predictions to provide enough resolution to distinguish strain-specific metabolic pathways [24].

In conclusion, predicted metabolic profiles associated to chlamydial infection may represent the starting point for more precisely designed future metabolomic studies, aiming to investigate the actual metabolic pathways characterizing *C. trachomatis* infection in the cervicovaginal microenvironment.

## 4. Materials and Methods

### 4.1. 16s rDNA Dataset

Cervicovaginal microbiota composition data were obtained from our previously published case-control study [2], which enrolled patients attending, for routine check-ups, the STIs Diagnostic Centre at Sant’Anna Hospital in Turin, as well as the Department of Gynecology, Obstetrics and Urology at the University “Sapienza” of Rome, Italy. The study population included a total of 39 women with *C. trachomatis* genital infection, and 99 matched healthy controls, ranging in age from 16 to 57 years. The details of the cohort have been elaborated on elsewhere [2].

Cervicovaginal swabs were collected from each patient and extracted DNAs were subjected to 16S rRNA gene amplicon library sequencing on the Illumina MiSeq platform via amplification of the V4 hypervariable region (primers 515F, 5′-GTGCCAGCMGCCGCGGTAA-3′, and 806R, 5′-GGACTACHVGGGTWTCTAAT-3′; Illumina Inc., San Diego, CA, USA), as previously described [2]. MiSeq paired-end reads were, then, re-analyzed by using the most updated plugins available in the software framework QIIME 2 (version 2022.2) [27]. Reads were subjected to demultiplexing and trimming of Illumina adaptor residuals using Illumina recommended parameter settings (Illumina MiSeq Reporter software, version 2.6). Primers were trimmed off the sequences using cutadapt (version 3.5) [28], and the resulting paired-end reads were then subjected to bioinformatic analysis using the QIIME 2 with the following steps: *i.* quality control; *ii.* denoising, aligning, and joining of paired-end reads, as well as identification and removal of chimeric sequences, via the plugin dada2 [29]; *iii.* Determination of alternative sequence variants (ASVs), individually unique and quality-filtered sequences analogous to operational taxonomic units, using the plugin dada2 [29,30].

The taxonomic assignment was performed by using a pre-trained out Bayes machine-learning classifier, trained to differentiate the taxa present in the 99% Greengenes reference sequences (version 13_8), trimmed to the V4 hypervariable region corresponding to the primers 515F and 806R, according to the methods described by Bokulich et al., 2018 with default parameters [31].

ASVs with only one sequence (singletons) and those not found more than 10 times in any sample were excluded from the downstream analysis to minimize artifacts. ASVs that could not be identified to a species level using the reference database, were searched using BLASTn and assigned to the deepest taxonomical level based on available published data. The microbial community of each patient was also assigned to their specific Community State Type (CST) following the approach described by Gajer et al. [12]. A flowchart of the bioinformatic pipeline is reported in Figure 6.

### 4.2. Phylogenetic Investigation of Communities by Reconstruction of Unobserved States (PICRUSt)

In order to identify potentially relevant metabolic pathways, metagenomic functional genes, and enzymatic reactions that may be associated with *C. trachomatis* genital infection, PICRUSt2 analysis was performed on the 16s rRNA gene dataset [21]. PICRUSt is a bioinformatic algorithm that uses evolutionary modeling to predict potential functions from 16s DNA sequence information and a reference genome database, and PICRUSt2 improves on the original method via the addition of larger database of gene families and reference genomes and interoperability with out-picking or denoising algorithm, enabling phenotypic predictions [21].

The ASV dataset was pre-processed to remove rare ASVs and singletons that could add noise to the analysis, as well as low-depth samples. PICRUSt2 pipeline (version 2.3.0_b) was subsequently run on the resulting 16s rDNA sequence data table as described in Douglas et al., 2020 [21]. The MetaCyc pathways, the highest-level predictions output by PICRUSt2, were used for investigating significant differences in the metabolic potential between *C. trachomatis*-positive and -negative women. The MetaCyc database [32] is an open-source alternative to the KEGG database, and its pathway abundances are calculated in PICRUSt2 through structured mapping of EC gene families to pathways. The MinPath algorithm was used to predict the pathway abundances based on the abundance of EC gene families.

### 4.3. Statistical Analysis

Shannon’s and Pielou’s evenness diversity indexes were used as measures of alpha diversity, whereas the Bray–Curtis index was used as a measure of beta diversity, for evaluating differences in richness and diversity of the predicted MetaCyc pathways between groups. A nonparametric *t*-test based on Monte Carlo permutations was used for alpha diversity comparisons, and Adonis was used for category comparisons of distance matrices, all calculated in QIIME 2 (version 2022.2) [27]. The relative abundances of MetaCyc pathways were expressed as means ± standard deviations (SD) and were analyzed by two independent samples unequal variance test (Welch’s *t*-test), calculated in the open-source STAMP software (version 2.1.3). Bonferroni correction was used to correct for multiple hypothesis testing when needed. The single or multiple inference significance levels was set at 5%.

## Figures and Tables

**Figure 1 ijms-23-15847-f001:**
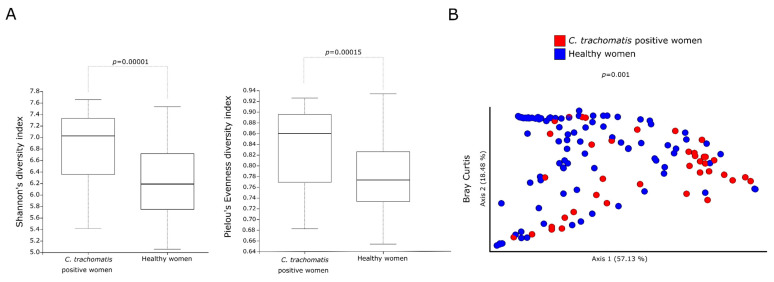
Alpha- and beta-diversity indexes of the predicted metabolic pathways between *C. trachomatis*-positive women and healthy controls. (**A**) Shannon’s and Pielou’s evenness indexes, used as a measure of alpha-diversity within groups; (**B**) Principal coordinate analysis of Bray–Curtis index, used as a measure of beta-diversity between groups. Each circle represents the predicted metabolic pathways in the cervicovaginal microbiota of each study subject.

**Figure 2 ijms-23-15847-f002:**
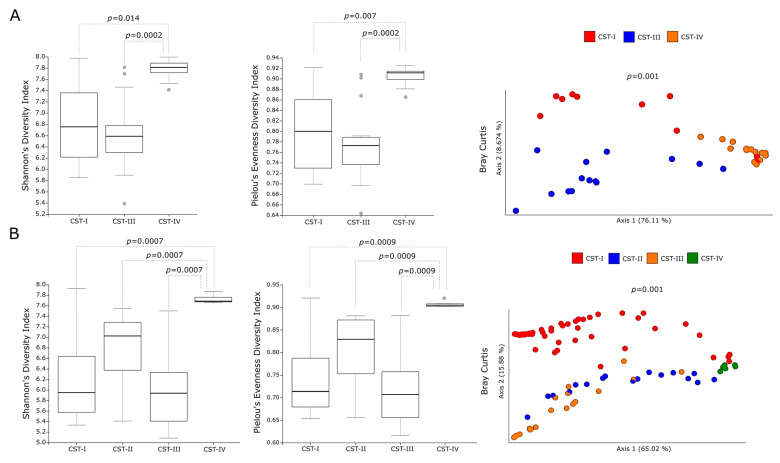
Alpha- and beta-diversity indexes of the predicted metabolic pathways in the different CSTs. Shannon’s, Pielou’s evenness, and Bray-Curtis indexes of the metabolic pathways in the different CSTs in (**A**) *C. trachomatis*-positive women; (**B**) healthy controls. The gray circles in the boxplots represent the outliers; each colored circle represents the predicted metabolic pathways in the cervicovaginal microbiota of each study subject.

**Figure 3 ijms-23-15847-f003:**
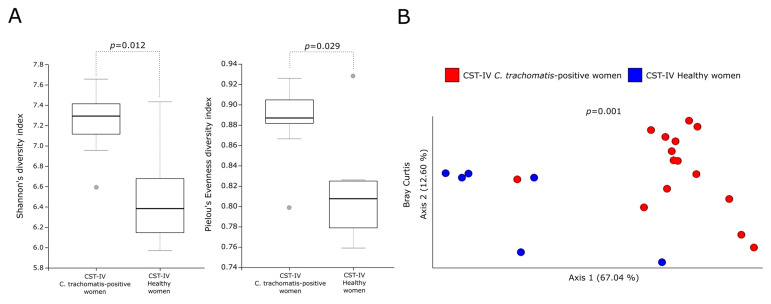
Alpha- and beta-diversity indexes of the predicted metabolic pathways in the CST-IV microbiota between *C. trachomatis*-positive women and healthy controls. (**A**) Shannon’s and Pielou’s evenness indexes; (**B**) Principal coordinate analysis of Bray-Curtis index. The gray circles in the boxplots represent the outliers; each colored circle represents the predicted metabolic pathways from each CST-IV microbiota.

**Figure 4 ijms-23-15847-f004:**
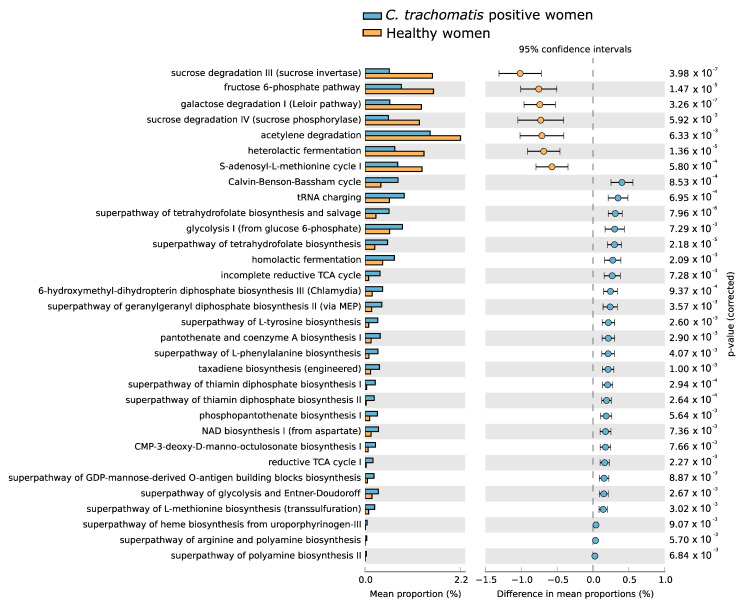
Differences in the composition of predicted metabolic pathways of the cervicovaginal microbiota between *C. trachomatis*-positive women and healthy controls. The relative abundance of each pathway was expressed as means ± standard deviations (SD), and statistical significance was calculated by Welch’s *t*-test.

**Figure 5 ijms-23-15847-f005:**
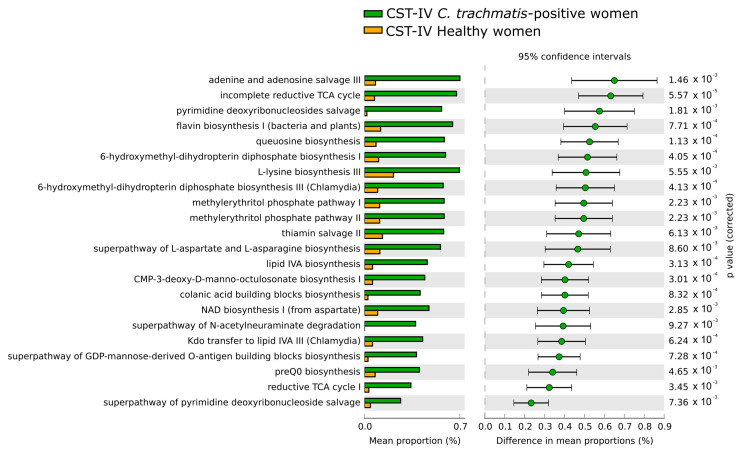
Differences in the composition of predicted metabolic pathways in the CST-IV microbiota between *C. trachomatis*-positive women and healthy controls. The relative abundance of each pathway was expressed as means ± standard deviations (SD), and statistical significance was calculated by Welch’s *t*-test.

**Figure 6 ijms-23-15847-f006:**
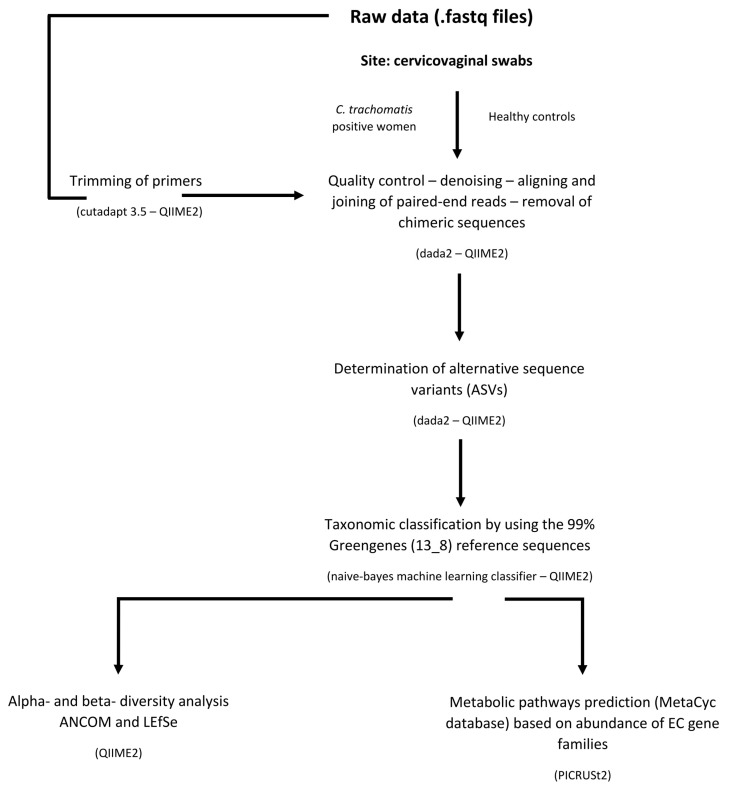
Flowchart of 16s rDNA paired-end Illumina reads bioinformatic pipeline.

## Data Availability

Raw sequences were deposited into the NCBI’s Sequence Read Archive (SRA) under accession number PRJNA509578.

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
