# Peer review of "In-Silico Functional Metabolic Pathways Associated to Chlamydia trachomatis Genital Infection"

_ijms, 2022, doi:10.3390/ijms232415847_

Round 1
Reviewer 1 Report
This study is well written about the in-silico functional metabolic pathways associated to Chlamydia trachomatis genital infection in relation to the different Community State types via the PICRUSt2 analysis. However, this reviewer could not find the novel concept in this manuscript. The authors had already reported the cervical microbial signatures in women with Chlamydia trachomatis infection (reference no. 2). In this study, the author had analyzed the prediction of metabolic pathway using PICRUSt2, which had not been proven with metabolome data. Borgogna et al., had already shown the metabolic pathways associated with C. trachomonatis with the metabolome data (reference no. 16). Although this manuscript had shown well-predicted metabolic profiles, this reviewer thought that the present manuscript had addressed the only narrow scope and that the further metabolome data should be analyzed and compared with the predicted data and the actually aquired metabolome data.
Reference
2. Filardo S, Di Pietro M, Tranquilli G, Latino MA, Recine N, Porpora MG, Sessa R. Selected Immunological Mediators and Cervical Microbial Signatures in Women with Chlamydia trachomatis Infection. mSystems. 2019 Jun 4;4(4):e00094-19. doi: 10.1128/mSystems.00094-19. PMID: 31164450; PMCID: PMC6550367.
16. Borgogna, JL.C., Shardell, M.D., Yeoman, C.J. et al. The association of Chlamydia trachomatis and Mycoplasma genitalium infection with the vaginal metabolome. Sci Rep 10, 3420 (2020). https://doi.org/10.1038/s41598-020-60179-z
Author Response
|
Reviewer 1’s Comments
|
Authors’ Answers
|
|
This study is well written about the in-silico functional metabolic pathways associated to Chlamydia trachomatis genital infection in relation to the different Community State types via the PICRUSt2 analysis. However, this reviewer could not find the novel concept in this manuscript. The authors had already reported the cervical microbial signatures in women with Chlamydia trachomatis infection (reference no. 2). In this study, the author had analyzed the prediction of metabolic pathway using PICRUSt2, which had not been proven with metabolome data. Borgogna et al., had already shown the metabolic pathways associated with C. trachomonatis with the metabolome data (reference no. 16). Although this manuscript had shown well-predicted metabolic profiles, this reviewer thought that the present manuscript had addressed the only narrow scope and that the further metabolome data should be analyzed and compared with the predicted data and the actually aquired metabolome data
|
We appreciate the Reviewer for their observations. The novelty of our paper consists in further exploiting the large body of information hidden within the 16s rDNA metagenomic data via powerful computational approaches. Indeed, our approach can be useful as a starting point for providing functional information that otherwise need untargeted metabolomic analysis, an expensive and time-consuming technique. As a matter of fact, the predicted metabolic pathways in our study, associated to C. trachomatis, are confirmed by data in the literature originating from untargeted metabolome data, as pointed out by the Reviewer.
|
Reviewer 2 Report
The authors have predicted metabolic pathways might represent the starting point for more precisely designed future metabolomic studies, aiming to investigate the actual metabolic pathways characterizing C. trachomatis genital infection in the cervicovaginal microenvironment.
The most surprising point is in method section where the authors have not even discussed anything about some of the critical things :
1. what was the k-mer length
2. was it blastn or blastx
3. what about the reads (total and assigned) and the sequencing quality? has the authors reanalyzed if not then must be mentioned in result section in summarized way.
4. Since the authors have used the previously published data so I could suggest that the authors could have also crosschecked the QIIME outcome with velvet because now velvet has been updated for 16sRNA/16sDNA and similarly diamond and megan could have used for the same. e.g., diamond could much better analyze the functions which the authrs have presented in Figure 4 and 5. I am not saying that it is mandatory to do that but will suggest to bring something in discussion section. The work presented by the authors sounds perfect and even the text written is well-written. Just for the improvement of the quality of this manuscript, I am suggesting this because I could clearly see the great potential for the people working in the field of metagenomics will be greatly benefited.
5. In result section I will request the authors to also discuss about the phylum, genus, and species and the figures could go into supplementary. Furthermore, the authors could also try to present the dendrogram with or without root.
6. Last suggestion will be to present a smart work flow in maintext so that the new audiences could easily reproduce or do the similar work. For sample I have attached my own workflow which I am mostly using. Please have a look over it.
Overall, I will say that this manuscript fulfils all the requirements for the next step of manuscript processing.

Author Response
|
Reviewer 2’s Comments
|
Authors’ Answers
|
|
The most surprising point is in method section where the authors have not even discussed anything about some of the critical things:
1. what was the k-mer length
|
We thank the Reviewer for the suggestions, and we added a detailed description of the methods used for the bioinformatic analysis of 16s rDNA sequences in Materials and Methods section (See page 11, lines 262-286). The k-mer length used was 7, as reported in the work by Bokulich et al., 2018 (Doi: 10.1186/s40168-018-0470-z).
|
|
2. was it blastn or blastx
|
The NCBI tool used was BLASTn.
|
|
3. what about the reads (total and assigned) and the sequencing quality? has the authors reanalyzed if not then must be mentioned in result section in summarized way.
|
We are sorry for the misunderstanding. We re-analysed the data with the most updated plugins available in the software framework QIIME2, and we added a detailed description of the bioinformatic pipeline in Materials and Methods section (see page 11, lines 262-286), as well as a summary of reads analysis results in Result section (see page 2, lines 80-94).
|
|
4. Since the authors have used the previously published data so I could suggest that the authors could have also crosschecked the QIIME outcome with velvet because now velvet has been updated for 16sRNA/16sDNA and similarly diamond and megan could have used for the same. e.g., diamond could much better analyze the functions which the authrs have presented in Figure 4 and 5. I am not saying that it is mandatory to do that but will suggest to bring something in discussion section. The work presented by the authors sounds perfect and even the text written is well-written. Just for the improvement of the quality of this manuscript, I am suggesting this because I could clearly see the great potential for the people working in the field of metagenomics will be greatly benefited.
|
We have utilized the most appropriate bioinformatic pipeline for the analysis of 16s rDNA amplicon sequence data based on amplification of V4 hypervariable region, as reported in the literature (Prodan A et al., 2020, doi: 10.1371/journal.pone.0227434). However we thank the Reviewer for the suggestion and in the future will be very interesting to compare different bioinformatic pipelines.
|
|
5. In result section I will request the authors to also discuss about the phylum, genus, and species and the figures could go into supplementary. Furthermore, the authors could also try to present the dendrogram with or without root.
|
We thank the Reviewer for the recommendation, and we added a paragraph concerning the phylum, genus and species in Result section (see page 2, lines 84-94).
|
|
6. Last suggestion will be to present a smart work flow in maintext so that the new audiences could easily reproduce or do the similar work. For sample I have attached my own workflow which I am mostly using. Please have a look over it.
|
We have added the workflow as supplementary figure S2.
|
Round 2
Reviewer 1 Report
In the discussion part, the authors should add the comparision of the predicted metabolic pathway with the previous reported GC-TOFMS and NMR data in detail.
Author Response
We thank the reviewer for the suggestion and we included the comparison of our predicted metabolic pathway with the previous reported GC-TOFMS and NMR data in detail (See Discussion section, page 10-11, lines 228-242).